# Use of Sedatives and Neuromuscular-Blocking Agents in Mechanically Ventilated Patients with COVID-19 ARDS

**DOI:** 10.3390/microorganisms9112393

**Published:** 2021-11-20

**Authors:** Amédée Ego, Lorenzo Peluso, Julie Gorham, Alberto Diosdado, Giovanni Restuccia, Jacques Creteur, Fabio Silvio Taccone

**Affiliations:** Department of Intensive Care, Erasme Hospital, Université Libre de Bruxelles (ULB), 1070 Brussels, Belgium; Lorenzo.peluso@erasme.ulb.ac.be (L.P.); Julie.Gorham@erasme.ulb.ac.be (J.G.); aldiosda@hotmail.es (A.D.); Giovanni.restuccia@gmail.com (G.R.); Jacques.creteur@erasme.ulb.ac.be (J.C.); Fabio.taccone@erasme.ulb.ac.be (F.S.T.)

**Keywords:** COVID-19, acute respiratory distress syndrome, sedation, neuromuscular-blocking agents

## Abstract

Objectives: To assess differences in the use of analgesics, sedatives and neuromuscular-blocking agents (NMBA) in patients with acute respiratory distress syndrome (ARDS) due to COVID-19 or other conditions. Methods: Retrospective observational cohort study, single-center tertiary Intensive Care Unit. COVID-19 patients with ARDS (March–May 2020) and non-COVID ARDS patients (2017–2020) on mechanical ventilation and receiving sedation for at least 48 h. Results: A total of 39 patients met the inclusion criteria in each group, with similar demographics at baseline. COVID-19 patients had a longer duration of MV (median 22 (IQRs 16–29) vs. 9 (6–18) days; *p* < 0.01), of sedatives administration (18 (11–22) vs. 5 (4–9) days; *p* < 0.01) and NMBA therapy (12 (9–16) vs. 3 (2–7) days; *p* < 0.01). During the first 7 days of sedation, compared to non-COVID patients, COVID patients received more frequently a combination of multiple sedative drugs (76.9% vs. 28.2%; *p* < 0.01) and a higher NMBA regimen (cisatracurium: 3.0 (2.1–3.7) vs. 1.3 (0.9–1.9) mg/kg/day; *p* < 0.01). Conclusions: The duration and consumption of sedatives and NMBA was significantly increased in patients with COVID-19 related ARDS than in non-COVID ARDS. Different sedation strategies and protocols might be needed in COVID-19 patients with ARDS, with potential implications on long-term complications and drugs availability.

## 1. Introduction

At the end of 2019, a new severe acute respiratory syndrome coronavirus 2 (SARS-CoV-2), responsible for the 2019 coronavirus disease (COVID-19), appeared and the World Health Organization (WHO) considered this infection as a “pandemic” on 11 March 2020. As of 3 November 2021, 246,303,023 cases and 4,994,160 deaths had been reported (https://www.who.int/emergencies/diseases/novel-coronavirus-2019/situation-reports (accessed on 3 November 2021)). Poor outcome factors have been described in several large multicenter cohorts, and relevant risk-factors for hospital mortality have been progressively identified, such as older age, male gender, the occurrence of renal failure, elevated D-dimer and C-reactive proteins, presence of immunosuppression, chronic respiratory failure, cardiovascular disease and/or diabetes [1,2]. Moreover, the presence of an acute respiratory distress syndrome requiring admission to the Intensive Care Unit (ICU) is associated with the highest mortality rate, often exceeding 40%, in particular among those patients requiring mechanical ventilation [3].

Early use of analgesic drugs and minimal sedation regimens are recommended for the management of patients suffering from acute respiratory distress syndrome (ARDS), which should favor awakening and early rehabilitation and reduce the risk of delirium and prolonged ventilation [4]. However, this approach might not be applied to severe ARDS patients, who might show symptoms of discomfort or ventilator asynchrony and therefore receive deep sedation and/or paralysis. Patients with COVID-19 ARDS have been shown to require high sedation regimens; in one retrospective study including 56 patients [5], the mean observed sedation period was more than 8 days and two or three sedative drugs were required in 49% and 13% of patients, respectively. Overall daily drug regimens were significantly higher than sedation regimens available in the literature for non-COVID ARDS, although the lack of a control group prevented any further analysis. Whether COVID-19 ARDS patients receive different sedation strategies to non-COVID ARDS patients remains poorly described. In one study [6], 92 patients with COVID-19 related ARDS required higher total median doses of propofol and were more likely receive intravenous lorazepam than 37 non-COVID ARDS patients; however, the two study groups were significantly unbalanced for several important variables and it remains difficult to conclude whether these differences in sedation requirement were due to COVID-19 or patients’ characteristics. Similarly, neuromuscular blocking agents (NMBA) have been largely used in COVID-19 patients with severe respiratory impairment on mechanical ventilation [7]; however, limited data exist to assess whether NMBA requirements would be higher in COVID-19 and other patients. 

The aim of this study was therefore to compare sedatives and NMBA requirements in these two ARDS patients’ populations.

## 2. Materials and Methods

### 2.1. Study Design

This retrospective cohort study was performed in the Department of Intensive Care of the Hôpital Erasme in Brussels, Belgium. The study protocol was approved by the local ethics committee (P2020/099), which waived the need for informed written consent because of its retrospective design and since all interventions were part of the standard patients’ care. 

### 2.2. Study Population

We enrolled patients admitted from March 10 (i.e., first admitted patient) to 30 June 2020 with the following inclusion criteria: (a) age of 18 years or older; (b) diagnosis of COVID-19 (i.e., on a positive polymerase chain reaction assay for SARS-CoV-2 on the nasopharyngeal swab and/or broncho-alveolar lavage specimens); (c) requiring mechanical ventilation; and (d) diagnosis of ARDS, according to standard criteria [8]. The non-COVID ARDS group included patients admitted in the ICU between January 2017 and September 2020, with the exclusion of those sedated for less than 48 h (i.e., to avoid selection bias due to less severe and short duration of respiratory impairment), with primary brain injury or acute liver failure (i.e., targeting the level of sedation based on clinical scales would be largely biased by the underlying disease). 

### 2.3. Data Collection and Sedation Management

We collected demographics, comorbidities and clinical characteristics on ICU admission for all patients. The use of propofol, midazolam, alpha-2 agonists (i.e., dexmedetomidine and clonidine) and/or ketamine was considered as “sedative”; the use of sufentanil, morphine and/or remifentanil as “analgesic”. Daily regimens were decided by the ICU physicians and dosing titrated by the nursing team based on the Nursing Instrument for the Communication of Sedation (NICS, −1 to 0 whenever possible) [9]. Protective lung ventilation (i.e., tidal volume of 6 mL/Kg ideal body weight, positive end-expiratory pressure > 5 cmH_2_O, plateau pressure < 30 cmH_2_O and driving pressure < 14 cmH_2_O) was applied in all patients, maintaining SpO_2_ between 90% and 94% and allowing moderate hypercapnia (PaCO_2_ 45–55 mmHg), provided the pH remained >7.25; in case of non-protective lung ventilation, sedation regimens were increased to obtain a NICS of −2. NMBAs included cisatracurium or rocuronium and were administered in case of PaO_2_/FiO_2_ < 150, asynchrony and/or persistent non-protective lung ventilation, despite deep sedation. All drugs were given as a continuous intravenous infusion; daily assessment of sedation and NMBA requirement was performed, with regard to ARDS severity and protective lung ventilation. Regimens of NMBAs were adapted to avoid overt asynchrony with the ventilator; train-of-four (TOF) measurements were not routinely implemented in all patients. Sedative, analgesic and NMBA strategies remained between the non-COVID and COVID-19 period.

Daily dose was calculated on the total amount of administered drugs over the first 7 days of ventilation and adjusted on the actual body weight, using the Patient Data Management Software (PDMS, PICIS Picis Critical Care Manager, Picis Inc., Wakefield, MA, USA). For opioids, the daily morphine equivalent consumption was computed [10]. Withdrawal from sedation and NMBAs was only initiated when respiratory mechanics improved (Delta Pressure < 14 cmH_2_O), positive end-expiratory pressure was ≤10 cmH_2_O and PaO_2_/FiO_2_ increased (i.e., >150 in supine position), although strategies were individualized daily at the medical round.

### 2.4. Outcome Assessment

The primary endpoint of this study was to the total amount of propofol, midazolam, and NMBA regimens between COVID and non-COVID patients. Secondary outcomes included: (a) the proportion of patients requiring more than one sedative drug; (b) duration of mechanical ventilation, sedation and ICU stay; and (c) daily morphine equivalent consumption.

### 2.5. Statistical Analysis

Statistical analyses were performed using the SPSS 28.0 for Macintosh (IBM, Armonk, NY, USA). Descriptive statistics were computed for all study variables. Discrete variables were expressed as count (percentage) and continuous variables as median (25th–75th percentiles), as appropriate. Demographics and clinical differences between groups were assessed using a chi-square test or Fisher’s exact test for categorical variable, and Student’s t-test or Mann–Whitney U-test for continuous variable, as appropriate. Because of the limited number of patients in both study cohorts, no further adjustment was provided to the analyses of main outcomes. A *p* < 0.05 was considered as statistically significant.

## 3. Results

A total of 39 COVID-19 and 39 non-COVID patients with ARDS were eligible for the study (Figure 1).

The main causes of non-COVID ARDS were bacterial infections, viral infections and polytrauma, as reported in Table 1.

Main demographics and comorbidities were similar between groups. The Simplified Acute Physiology Score (SAPS) III and Sequential Organ Failure Assessment (SOFA) scores on day 1 were higher in the non-COVID-19 group (56 (50–67) vs. 66 (55–78); *p* = 0.02 and 7 (4–9) vs. 11 (7–13); *p* < 0.01 respectively—Table 2), as the occurrence of acute kidney injury (AKI) on admission.

During the study period, seven patients (18%) were treated with venous-venous extracorporeal membrane oxygenation (VV-ECMO) and two (5%) with renal replacement therapy in the COVID group when compared to 11 (28%; *p* = 0.42) and 12 (31%; *p* < 0.01) in the non-COVID group, respectively. ICU survival was similar between groups (54% vs. 59%, *p* = 0.82). No patient had circulatory failure requiring high doses of catecholamines (i.e., norepinephrine > 1.0 mcg/Kg.min).

Median duration of MV, of sedation and ICU stay were longer in the COVID-19 group (22 (16–29) vs. 9 (6–18) days; *p* < 0.01—18 (11–22) vs. 5 (4–9) days; *p* < 0.01—27 (19–32) vs. 14 (9–22) days; *p* < 0.01, respectively) than the non-COVID group. In the COVID-19 group, a higher proportion of patients (30/39, 76.9% vs. 11/39, 28.2%; *p* < 0.01) received a combination of sedatives when compared to others (Figure 2). 

The median daily regimen of propofol and midazolam were similar between groups (64 (47–77) vs. 52.4 (44–68) mg/kg/day, *p* = 0.15, and 1.3 (1.0–1.6) vs. 1.2 (1.1–1.4) mg/kg/day; *p* = 0.42, respectively). There were more patients who received the administration of ketamine (11/39, 28% vs. 3/39, 8%, *p* = 0.04), although daily regimens were similar (17.0 (8.9–22.0) vs. 33.7 (25.2–38.3) mg/kg/day; *p* = 0.13). Twelve COVID-19 (39%) and eight non-COVID (21%) patients received the administration of alpha-2-agonists, with a similar daily regimen (27.4 (18.7–38.6) vs. 38.6 (28.2–44.7) mcg/kg/day; *p* = 0.65) (Figure 3, Table 3). The daily morphine equivalent consumption was lower in the COVID-19 than non-COVID group (0.9 (0.5–1.5) vs. 1.3 (0.8–2.0) mg/kg/day; *p* = 0.04). All COVID and 37/39 non-COVID (95%) patients received NMBAs (*p* = 0.49); however, the duration of therapy was longer in the COVID-19 groups (12 (9–16) vs. 3 (2–7) days; *p* < 0.01). The daily regimen of cisatracurium was higher in the COVID-19 group (2.3 (2.1–3.7) vs. 1.3 (0.9–1.9) mg/kg/day; *p* < 0.01). Only one patient in the non-COVID group received rocuronium, while five patients received this drug in the COVID-19 group for several days, either due to tachyphylaxis or shortage of cisatracurium (Table 3).

## 4. Discussion

In this study, we observed that COVID-19 ARDS patients received more frequently a combination of multiple sedative drugs than non-COVID patients to reach defined sedation targets. Also, increased daily regimens of NMBAs were received in COVID-19 when compared to non-COVID patients with similar characteristics to allow lung protective ventilation and reduce asynchronies. The duration of sedatives and NMBA administration was longer in such patients, resulting in a longer duration of mechanical ventilation and ICU stay. Opioid consumption was significantly lower in the COVID-19 group. 

These findings were consistent with three previous studies [6,11,12], which reported an increased need of sedative drugs in COVID-19 ARDS patients when compared to non-COVID patients; however, study groups were not comparable (i.e., different characteristics and severity; unmatched cohorts) and the control non-COVID group could not reflect the actual management of ARDS patients on MV [11]. Our findings might have some important implications for clinical practice: (a) sedation protocols, which are often based on one single drug, should be revised in COVID-19 patients with ARDS on MV (i.e., more combination therapy) to obtain the same level of sedation than in non-COVID ARDS patients; (b) if NMBAs are more frequently administered, the risk for long-term complications (i.e., ICU-acquired weakness) should be carefully evaluated; (c) the high afflux of patients and the requirement for increased doses and prolonged duration of sedatives and NMBAs might result in limited drug availability and might result in significant organizational issues for drug supply and stockage during the pandemic.

How can these observations be explained when clinical guidelines recommend light sedation in patients on MV? First, management of ventilated patients with light sedation requires more attention and is associated with higher workload, while human resources were limited during the pandemic. However, sedation targets remained similar in COVID-19 and non-COVID cohorts and the number of nurses and doctors was significantly increased during the COVID-19 pandemic in our ICU, to allow a more accurate care of these patients. Second, overuse of sedatives could be related to a more severe hypoxemia, which would result in deep sedation to provide advanced ventilatory support (i.e., prone positioning or ECMO) or to younger age; however, demographics, severity of disease and of hypoxemia were similar between groups in our study. Third, the profound and persistent inflammatory state associated with COVID-19 might increase drug metabolism, thus requiring higher daily regimens; however, we did not specifically analyze drug concentrations in this cohort. Fourth, a more significant respiratory drive, triggered by thoracic receptors activated by severe pulmonary inflammation, could be present during COVID-19 and therefore require higher doses of sedatives and NMBAs to avoid spontaneous breathing and allow protective ventilation [13,14]. Fifth, as there was a lot of concern about the risk of aerosolization at the beginning of the pandemic, it is possible that high doses of sedatives and NMBAs have been used by ICU physicians and nurses to prevent potential contamination during high-risk maneuvers (i.e., endotracheal aspiration, mobilization). However, the sedation targets remained similar over time and no specific recommendations on aiming to a deeper level of sedation in COVID-19 patients were introduced in clinical practice. Sixth, prolonged periods of sedation may lead to drug tolerance and tachyphylaxis; however, we specifically focused on the first week of ventilation, when this phenomenon should be less frequent. The prolonged duration and intensity of sedative and NMBAs administration could also trigger the occurrence of delirium or ICU-acquired weakness, which could result in work overload for ICU teams and poor outcome. In one study including 2088 patients [15], infusion with sedatives while on mechanical ventilation was recorded in 64% of them and the use of benzodiazepines and opioids were some of the independent predictors of delirium. In another study [16], critical illness polyneuropathy and myopathy was observed in 11 out of 111 patients, however no specific risk-factors were reported. Importantly, we did not specifically assess the presence of ICU-acquired weakness in this study cohort. Finally, the lower daily regimens of opioids is difficult to interpret because it is possible that the higher daily sedative and NMBA regimens contributed to alter the assessment of nociception (i.e., self-reported scales or behavioral scales), which resulted in lower daily regimens than non-COVID patients.

The limitations of our study were: (a) the lack of monitoring of sedation (i.e., using electroencephalography rather than sedation scales) and depth of paralysis (i.e., using NMBA monitoring) to adjust daily regimens; however, there are no recommendations on the routine use and validity of such techniques in ICU patients; (b) the possibility that bolus injections could not be adequately recorded into the PDMS, thus contributing to the underestimation of daily drug regimens; (c) limited study cohort, with unmatched control group; (d) single center study, although this reduced the heterogeneity in sedation practices (i.e., same medical staff, same sedation protocols with fixed targets) and allowed an adequate assessment of the effects of COVID-19 on drug prescription; (e) data related only to the first wave, while sedation practices may have changed over time; (f) the lack of assessment of other potential sedatives-related complications, such as delirium, muscular weakness and ICU-acquired infections; and (g) the lack of assessment of residual renal function over the observation time, which might have influenced drug accumulation and sedative effects.

## 5. Conclusions

In conclusion, COVID-19 patients with ARDS received higher daily regimens including multiple drugs and prolonged administration of sedatives, as well as prolonged use of NMBA than non-COVID patients. Optimization of these interventions, using specific therapeutic and monitoring protocols, needs to be further assessed in COVID-19 patients.

## Figures and Tables

**Figure 1 microorganisms-09-02393-f001:**
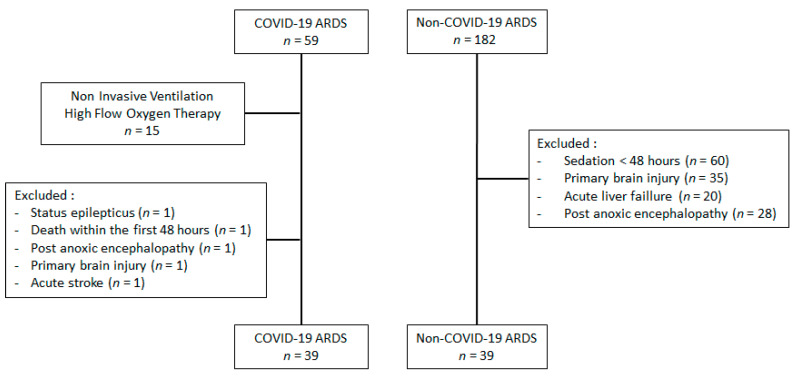
Flow chart of the study.

**Figure 2 microorganisms-09-02393-f002:**
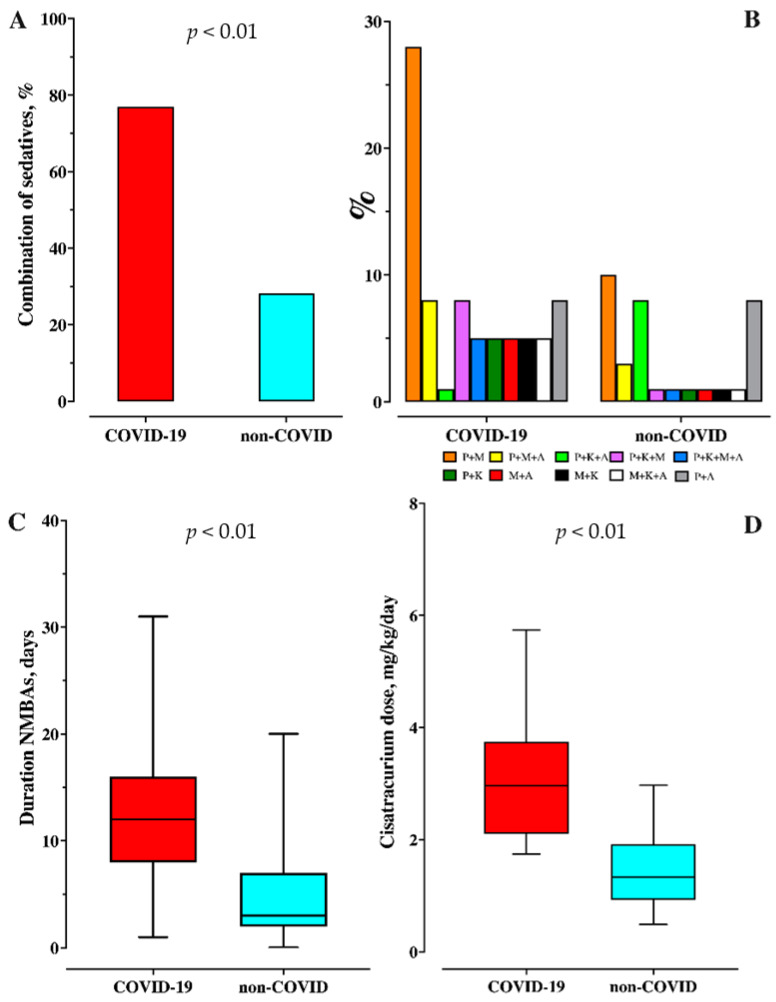
Combination of sedatives, analgesics and neuromuscular-blocking agents. (**A**): Proportion of patients receiving a combination of sedatives. (**B**): Description of the combination of sedatives. (**C**): Duration of administration of Neuromuscular blocking agents (NMBA). (**D**): Cisatracurium administration regimen. P+M = Propofol + midazolam; P+M+A = Propofol + midazolam + alpha-2-agonist; P+K+A = Propofol + ketamine + alpha-2-agonist; P+K+M = Propofol + ketamine + midazolam; P+K+M+A = Propofol + ketamine + midazolam + alpha-2-agonist; Propofol + ketamine; M+A = Midazolam + alpha-2-agonist; M+K = Midazolam + ketamine; M+K+A = Midazolam + ketamine + alpha-2-agonist; P+A = Propofol + alpha-2-agonist.

**Figure 3 microorganisms-09-02393-f003:**
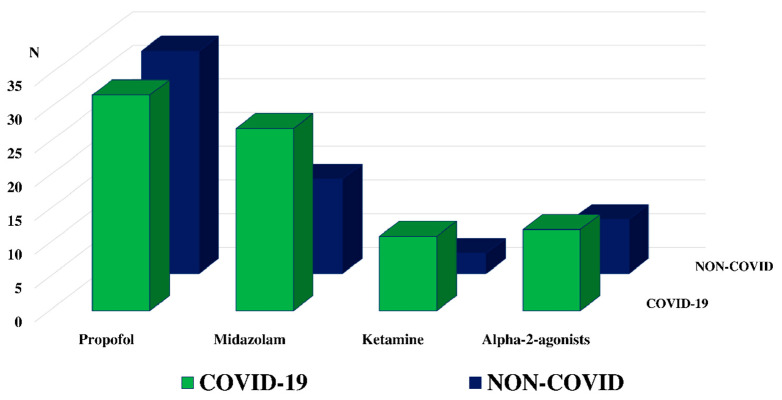
Distribution of sedative use according to ARDS COVID-19 versus non-COVID-19 ARDS.

**Table 1 microorganisms-09-02393-t001:** Causes of ARDS in non-COVID-19 patients.

Causes of Non-COVID-19 ARDS		*n* = 39
**Pneumonia**		
*Bacterial*	Escherichia Coli	6
	Enterobacter Cloacae	4
	Klebsiella pneumoniae	2
	Pseudomonas Aeroginosa	2
	Haemophilus influenzae	1
	Enterobacter Aerogenes	1
	Enterococcus faecalis	1
	Corynebacterium stratium	1
	Streptococcus pneumoniae	1
	Hafnia alvei	1
*Viral*	Influenza A	5
	Influenza B	1
	Cytomegalovirus	1
*Fungic*	Pneumocystis jirovecii	1
**Contusion**		
	Polytrauma	4
**Extra-pulmonary**		
	Mesenteric ischemia	2
	Endocarditis	1
**Others**		
	Alveolar hemorrhage	1
	Side effect of CAR-T cells therapy	1
	Autoimmune	1
	Peritonitis	1

**Table 2 microorganisms-09-02393-t002:** Characteristics of the study cohorts. Data are expressed as counts (percentage) and median (interquartile range). Legend: BMI = Body Mass Index; COPD = Chronic obstructive pulmonary disease; NYHA = New York Heart Association score; SAPS 3 = Simplified acute physiology score; SOFA = Sepsis-related Organ Failure; MV = Mechanical Ventilation; NMBA = Neuromuscular blocking agents; ECMO = Extracorporeal membrane oxygenation; RRT = Renal replacement therapy.

	COVID-19 (*n* = 39)	Non-COVID-19 (*n* = 39)	*p* Value
Age, (years)	59 (53–65)	52 (45–68)	0.07
Male Gender, *n* (%)	29 (74)	24 (62)	0.33
BMI	29.4 (25–32)	26 (23–31)	0.06
Comorbidities			
COPD/Asthma, *n* (%)	9 (23)	13 (33)	0.45
Arterial hypertension, *n* (%)	18 (13)	14 (14)	0.59
Admission Serum Creatinine (mg/dL)	1.06 (0.76–1.44)	0.95 (0.7–1.45)	0.37
Heart failure (NYHA I-II), *n* (%)	4 (10)	3 (8)	1.00
Tobacco	4 (10)	9 (23)	0.22
Alcohol	2 (5)	1 (3)	1.00
Severity Scores			
SAPS 3	56 (50–67)	66 (55–78)	0.02
SOFA day 1	7 (4–9)	11 (7–13)	<0.01
PaO2/FiO2 24 h post MV	141 (112–200)	131 (105–178)	0.11
Duration therapies			
Mechanical Ventilation (days)	22 (16–29)	9 (6–18)	<0.01
Sedation (days)	18 (11–22)	5 (4–9)	<0.01
NMBA use, *n* (%)	39 (100)	37 (95)	0.49
NMBA (days)	12 (8.5–16)	3 (2–7)	<0.01
ECMO V-V per 7 days study, *n* (%)	7 (18)	11 (28)	0.42
RRT per 7 days study, *n* (%)	2 (5)	12 (31)	<0.01
ICU stay (days)	27 (19–32)	25 (14–41)	0.23
ICU survival, *n* (%)	21 (54)	23 (59)	0.82

**Table 3 microorganisms-09-02393-t003:** Drug consumption. Data are expressed as counts (percentage) and median (interquartile range). Legend: NMBA = Neuromuscular blocking agents.

	COVID-19 (*n* = 39)	Non-COVID-19 (*n* = 39)	*p* Value
Propofol, *n*, (%)	32 (82)	33 (85)	1
Propofol, mg/kg/day	64 (47–77)	52.4 (44–68)	0.15
Midazolam, *n* (%)	27 (69)	14 (36)	0.01
Midazolam, mg/kg/day	17 (9–22)	34 (25–38)	0.13
Ketamine, *n* (%)	11 (28)	3 (8)	0.04
Morphine, *n* (%)	32 (82)	38 (97)	0.06
Morphine equivalent, mg/kg/day	0.9 (0.5–1.5)	1.3 (0.8–2.0)	0.04
Alpha2-agonist, *n* (%)	12 (39)	8 (21)	0.44
Clonidine equivalent, mcg/kg/day	27.4 (19–36)	39 (28–45)	0.65
Cisatracurium, mg/kg/day	2.9 (2.1–3.7)	1.3 (0.9–1.9)	< 0.01

## Data Availability

The data presented in this study are available on request from the corresponding author.

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
