# Peer review of "Use of Sedatives and Neuromuscular-Blocking Agents in Mechanically Ventilated Patients with COVID-19 ARDS"

_microorganisms, 2021, doi:10.3390/microorganisms9112393_

Round 1
Reviewer 1 Report
I would like to congratulate the authors for the manuscript entitled: “Use of Sedatives and Neuromuscular-Blocking agents in Mechanically Ventilated patients with COVID-19 ARDS”. The manuscript is well written and easy to follow. Furthermore, conclusions in this single centre study are solid and in line with previous studies.
I would like to make just two minor comments:
Comment 1, Is it possible to obtain the frequencies of non-COVID ARDS causes as a table in the supplemental material or in the manuscript itself? As mentioned, most of the non-COVID patients had bacterial pneumonia, but it would be interesting to know the included diagnosis. The table would help to understand the control group.
Comment 2, In figure 2, there is a minor error when explaining acronyms. The second combination acronym after P+M, should be P+M+A = Propofol + midazolam + alpha-2-agonist instead of P+M+K.
Comment 3, In the discussion section, the authors explain why characteristics and sedation targets were similar between both groups. They also describe an increased staff taking care of the patients during COVID19 pandemic, being easier to monitor sedation and drug doses. Thus, why do authors think there is a higher use of sedative drugs and NMBA in COVID19? Is this due to the severe inflammation produced by COVID19 and a specific effect of this particular disease? It has to do with the first wave of COVID19 pandemic and the overwhelmed health system? Have authors checked what happened in the next waves? As explained, if it is a particular finding in this disease, sedation protocols should be revised.
Author Response
- I would like to congratulate the authors for the manuscript entitled: “Use of Sedatives and Neuromuscular-Blocking agents in Mechanically Ventilated patients with COVID-19 ARDS”. The manuscript is well written and easy to follow. Furthermore, conclusions in this single centre study are solid and in line with previous studies.
Authors’ response: We thank the reviewer for the nice comment.
- Is it possible to obtain the frequencies of non-COVID ARDS causes as a table in the supplemental material or in the manuscript itself? As mentioned, most of the non-COVID patients had bacterial pneumonia, but it would be interesting to know the included diagnosis. The table would help to understand the control group.
Authors’ response: The table has been added to the manuscript, notably with microbiology findings, as requested.
- In figure 2, there is a minor error when explaining acronyms. The second combination acronym after P+M, should be P+M+A = Propofol + midazolam + alpha-2-agonist instead of P+M+K.
Authors’ response: This has been modified, accordingly.
- In the discussion section, the authors explain why characteristics and sedation targets were similar between both groups. They also describe an increased staff taking care of the patients during COVID19 pandemic, being easier to monitor sedation and drug doses. Thus, why do authors think there is a higher use of sedative drugs and NMBA in COVID19? Is this due to the severe inflammation produced by COVID19 and a specific effect of this particular disease? It has to do with the first wave of COVID19 pandemic and the overwhelmed health system? Have authors checked what happened in the next waves? As explained, if it is a particular finding in this disease, sedation protocols should be revised.
Authors’ response: The reviewer has raised an important issue. Whether the higher and longer use of sedatives was due to one or more specific reasons, it remains unknown. Our work could not identify a specific reason, due to retrospective data collection. We did not assess the consumption on the following COVID-19 related ICU waves, and this has been addressed as a limitation, accordingly.
Reviewer 2 Report
Dear Authors,
The manuscript is well organized and written.
Author Response
- The manuscript is well organized and written.
Authors’ response: We thank the reviewer for the nice comment.
Reviewer 3 Report
I read with great interest the paper. I find it very well wrote and with interesting point of view.I dea research and presentation is high quality. It is not easy give some suggestions. Below only some minor comments:
- Introduction: updata data on SARS CoV2 wordwilde. Furthermore, introduce the comorbidity that expose the patients to worst outcome (Common cardiovascular risk factors and in-hospital mortality in 3,894 patients with COVID-19: survival analysis and machine learning-based findings from the multicentre Italian CORIST Study. Nutr Metab Cardiovasc Dis. 2020 Oct 30;30(11):1899-1913.)
- Methods and results: very clear presented
- Discussion: discuss also the role of therapies on outcome (es Heparin in COVID-19 Patients Is Associated with Reduced In-Hospital Mortality: The Multicenter Italian CORIST Study. Thromb Haemost. 2021 Aug;121(8):1054-1065.) and propose some action that came from your very interesting results
Author Response
- The manuscript is well organized and written. I read with great interest the paper. I find it very well wrote and with interesting point of view. I dea research and presentation is high quality. It is not easy give some suggestions.
Authors’ response: We thank the reviewer for the nice comment.
- Introduction: update data on SARS CoV2 word-wilde. Furthermore, introduce the comorbidity that expose the patients to worst outcome (Common cardiovascular risk factors and in-hospital mortality in 3,894 patients with COVID-19: survival analysis and machine learning-based findings from the multicentre Italian CORIST Study. Nutr Metab Cardiovasc Dis. 2020 Oct 30;30(11):1899-1913.)
Authors’ response: We added up-to-date epidemiological data according to the WHO, as well as the co-morbid factors influencing the poor outcome, as requested.
3. Methods and results: very clear presented
Authors’ response: Many thanks for this comment.
4. Discussion: discuss also the role of therapies on outcome (es Heparin in COVID-19 Patients Is Associated with Reduced In-Hospital Mortality: The Multicenter Italian CORIST Study. Thromb Haemost. 2021 Aug;121(8):1054-1065.) and propose some action that came from your very interesting results
Authors’ response: We thank the reviewer for this comment. As the scope of the study was to describe sedation practices, we did not specifically address the role of additional therapies of COVID-19, as these might be significantly different than non-COVID patients.
Reviewer 4 Report
The Authors present a retrospective comparison of sedative and NMB use in Covid and non-Covid Ards patients. While the topic is of interest, I do have some concerns about methodology and description.
- Introduction can be improved, in number of citations for the background from where the paper takes place and in writing.
- The demographic differences, in particular the different number of patients with AKI, should be discussed further
- In Figure 2, B, it is difficult to understand and compare different drug regimens; I would suggest 2 paired colums "Covid-nonCovid" for each drug regimen, in different colours
- In the Discussion, you state "Opioid consumption was significantly lower in the COVID-19 group" and just after this you contradict yourself stating "These findings were consistent with three previous studies [3,8,9], which reported an increased need of analgesic and sedative drugs in COVID-19 ARDS patients"
- How do you discuss your results with the number of patients with AKI in a context where drug clearance and drug consumption could be affected by it?
- All your suggested explanations for bigger sedative and NMB use in Covid are interesting but a major limitation of the study, as you state, is that no monitor of sedation or NM blockade has been applied thus leaving the doubt of appropriateness of drug dosing, in light also of a situation where human resources (health care workers) were under pressure due to the epidemiological situation
- The discussion about delirium and critical illness mio-polineuropathy is inappropriate since this evenience has not been searched in the cohort
- All the limitations you state are correct, leaving however small space for a correct discussion.
All the discussion requires re-writing. It is possible that changing the objective the paper can be accepted. In this form it gives the reader very few new information.
Author Response
1. Introduction can be improved, in number of citations for the background from where the paper takes place and in writing.
Authors’ response: We have improved the introduction, adding up-to-date demographics and poor outcome factors.
2. The demographic differences, in particular the different number of patients with AKI, should be discussed further
Authors’ response: We thank the reviewer for this comment. We have added the serum creatinine to the Table describing patients’ population, to have a more complete view of the patients. Importantly, we did not superficially assess the occurrence of AKI (at any stage) during the ICU stay.
3. In Figure 2, B, it is difficult to understand and compare different drug regimens; I would suggest 2 paired columns "Covid-nonCovid" for each drug regimen, in different colors
Authors’ response: We have added a table showing the use of different sedatives according to the two groups.
4. In the Discussion, you state "Opioid consumption was significantly lower in the COVID-19 group" and just after this you contradict yourself stating "These findings were consistent with three previous studies [3,8,9], which reported an increased need of analgesic and sedative drugs in COVID-19 ARDS patients"
Authors’ response : We agree with the reviewer – discussion has been modified accordingly.
5. How do you discuss your results with the number of patients with AKI in a context where drug clearance and drug consumption could be affected by it?
Authors’ response : We thank the reviewer for this comment. It is difficult to make a complete statement on this issue, as AKI was evaluated on admission, but we did not specifically assess the occurrence of AKI during the ICU stay. Importantly, the use of continuous renal replacement therapy over time was higher in the non-COVID patients (i.e. higher occurrence of KDIGO III AKI during the observation period ?). Moreover, if AKI would have been higher in COVID-19 patients (i.e. higher drug accumulation), this would have resulted in lower, and not higher, sedation requirement to obtain target sedation end-point.
6. All your suggested explanations for bigger sedative and NMB use in Covid are interesting but a major limitation of the study, as you state, is that no monitor of sedation or NM blockade has been applied thus leaving the doubt of appropriateness of drug dosing, in light also of a situation where human resources (health care workers) were under pressure due to the epidemiological situation
Authors’ response: The use of sedation depth monitoring, analgesia and NMBA have not been validated for intensive care patients and are not part of international recommendations. Therefore, we do not use them routinely in intensive care. Only clinical scales have been validated in intensive care. As this is an observational study, we do not have these data.
7. The discussion about delirium and critical illness myo-polyneuropathy is inappropriate since this evidence has not been searched in the cohort
Author’s response: We agree that these data were not specifically collected, however these complications remain relevant and highly related to the use of NMBAs and sedatives and it remains important to discuss them in the manuscript.
8. All the limitations you state are correct, leaving however small space for a correct discussion.
Authors’ response: We thank the reviewer. Limitations have been updated according to all reviewers’ criticisms.
Round 2
Reviewer 4 Report
Dear Authors,
I thank you for the replies to my previous remarks. One little new remark: in Table 2, in the explanation of table, the term AKI is explained but it has been deleted from the list of analyzed items. Beside that, is is suitable for publication.
Good luck!
Author Response
We thank the reviewer for this observation. This has been modified, as requested.